# Next Generation Sequencing after Invasive Prenatal Testing in Fetuses with Congenital Malformations: Prenatal or Neonatal Investigation

**DOI:** 10.3390/genes13091517

**Published:** 2022-08-24

**Authors:** Alexandra Emms, James Castleman, Stephanie Allen, Denise Williams, Esther Kinning, Mark Kilby

**Affiliations:** 1West Midlands Fetal Medicine Centre, Birmingham Women’s and Children’s NHS Foundation Trust, Birmingham B15 2TG, UK; 2West Midlands Regional Genetics Laboratory, Birmingham Women’s and Children’s NHS Foundation Trust, Mindelsohn Way, Edgbaston, Birmingham B15 2TG, UK; 3Clinical Genetics Service, Birmingham Women’s and Children’s NHS Foundation Trust, Birmingham B15 2TG, UK; 4Institute of Metabolism and Systems Research, College of Medical and Dental Sciences, University of Birmingham, Birmingham B15 2TT, UK

**Keywords:** fetus, neonate, exome, genome, sequencing

## Abstract

Congenital malformations diagnosed by ultrasound screening complicate 3–5% of pregnancies and many of these have an underlying genetic cause. Approximately 40% of prenatally diagnosed fetal malformations are associated with aneuploidy or copy number variants, detected by conventional karyotyping, QF-PCR and microarray techniques, however monogenic disorders are not diagnosed by these tests. Next generation sequencing as a secondary prenatal genetic test offers additional diagnostic yield for congenital abnormalities deemed to be potentially associated with an underlying genetic aetiology, as demonstrated by two large cohorts: the ‘Prenatal assessment of genomes and exomes’ (PAGE) study and ‘Whole-exome sequencing in the evaluation of fetal structural anomalies: a prospective cohort study’ performed at Columbia University in the US. These were large and prospective studies but relatively ‘unselected’ congenital malformations, with little Clinical Genetics input to the pre-test selection process. This review focuses on the incremental yield of next generation sequencing in single system congenital malformations, using evidence from the PAGE, Columbia and subsequent cohorts, with particularly high yields in those fetuses with cardiac and neurological anomalies, large nuchal translucency and non-immune fetal hydrops (of unknown aetiology). The total additional yield gained by exome sequencing in congenital heart disease was 12.7%, for neurological malformations 13.8%, 13.1% in increased nuchal translucency and 29% in non-immune fetal hydrops. This demonstrates significant incremental yield with exome sequencing in single-system anomalies and supports next generation sequencing as a secondary genetic test in routine clinical care of fetuses with congenital abnormalities.

## 1. Introduction

Congenital malformations diagnosed using prenatal ultrasound screening complicate approximately 3–5% of pregnancies. Classic teaching is that associated monogenic aetiology is relatively rare as an association (~5%). However, aneuploidy, structural chromosomal rearrangements (large or submicroscopic) and copy number variants (CNVs) are associated with up to 40% of congenital malformations [1]. These are diagnosed using conventional G-band karyotyping, QF-PCR and microarray analysis. These tests usually do not diagnose underlying monogenic diseases. The next step in the diagnosis of monogenic disorders prenatally is to use next generation sequencing (NGS) techniques. In this context, NGS utilises test DNA from the fetus obtained by amniocentesis, chorionic villous or fetal blood sampling and involves trio (both parents and fetus) exome sequencing (ES) and potentially whole genome sequencing (WGS). There have been two relatively large cohort studies: ‘Prenatal assessment of genomes and exomes’ (PAGE) (Lord et al. [2]) and ‘Whole-exome sequencing in the evaluation of ‘unselected’ fetal structural anomalies: a prospective cohort study performed at Columbia University’ by Petrovski et al. [3] which have shown enhanced diagnostic yield with exome sequencing when conventional testing is non-informative. In these studies, Clinical Geneticists were not necessarily involved in the selection of cases. Clinical diagnostic yield was relatively low at 10–12% overall. Selection of cases using a ‘pre-test’ multidisciplinary team (including a Clinical Geneticist) and careful ‘targeting’ of specific fetal phenotypes may significantly increase diagnostic rate to over 30% [4,5,6].

Achieving a genetic diagnosis in fetal life may improve clinical care by informing individualised genetic counselling for the current and future pregnancies, allowing discussion regarding prognosis and treatment with the appropriate paediatric specialists, and may support parents in decision making regarding termination of pregnancy. In England, from October 2020, the National Health Service (NHS) commissioned a rapid prenatal exome sequencing pathway which includes trio exome sequencing, and then analysis using a nationally agreed panel of genes. This “fetal anomaly gene panel” is updated regularly, and currently includes approximately 1200 genes (R21), which have an evidence-base to link them to fetal phenotypes [7,8]. Although at significant cost, the introduction of a national clinical pathway for prenatal testing allows the prospective detection of rare genetic disorders associated with structural malformations. Next generation sequencing is also utilised postnatally for neonates and infants requiring intensive care and thought to have a potential monogenic disease, via the R14 pathway (trio whole exome sequencing with gene agnostic analysis) in the NHS National Genomic Test Directory or via the R27 pathway (trio whole genome sequencing with gene panel-based analysis) in case of a pregnancy loss (miscarriage, stillbirth or early neonatal death) [4,9]. These current pathways all require DNA to be obtained directly from the fetus or neonate (placental biopsy, amniocytes, blood or tissue sample).

## 2. Prenatal Imaging for Diagnosis of Fetal Structural Differences

Prenatal ultrasound imaging as a screening tool for structural anomalies is well established as part of routine antenatal care, with specific guidelines for a standardised approach to imaging first released internationally by the International Society of Ultrasound in Obstetrics and Gynecology (ISUOG) in 2010 [10]. Adopted by the UK, the NHS Fetal Anomaly Screening Programme (FASP) [11] provides structured guidance for the evaluation of fetal anatomy as part of the antenatal screening pathway. Structural anomalies are found in approximately 3–5% of pregnancies [1,12] undergoing prenatal ultrasound screening and these patients are referred to Fetal Medicine specialists, often in tertiary centres, for further prenatal imaging and review. There are many causes of fetal anomaly, for example fetal infection, or teratogenicity related to maternal medications. A detailed Fetal Medicine ultrasound aims to identify and characterise structural differences using dysmorphology principles [13] to describe the prenatal phenotype (which may evolve as the gestation advances) and establish a differential diagnosis. If this differential diagnosis includes a chromosome difference or monogenic disorder, then genetic testing may be offered through amniocentesis or chorionic villus sampling. Diagnostic imaging only tells part of the story, and therefore Fetal Medicine specialists will seek more information about the aetiology of the ultrasound phenotype in order to counsel parents about the prognosis and refer them appropriately for ongoing care. There are many subtle phenotypic features in monogenic disease that cannot be identified using ultrasound, despite advances in ultrasonography techniques. There can be variations in phenotypic expression at different gestations, which may limit the ability to achieve a diagnosis. In utero magnetic resonance imaging is a useful adjunct to define the prenatal phenotype (in specific cases) [14], and tests such as maternal viral serology can exclude other aetiologies. Clinical suspicion of monogenic disease is increased in cases with close relative partnerships [15], relevant obstetric or family history and multi-system structural differences. Specific targeting of fetal phenotypes is discussed later in this article.

## 3. Clinical Genetics

Multi-disciplinary working between Fetal Medicine and Clinical Genetics is key to achieving not only a prospective diagnosis but also to providing the necessary pre-test counselling that parents undergoing prenatal genetic testing for fetal anomalies require. This is most effective in the form of combined clinics. Over a 10-year period, a joint fetal medicine/genetic clinic in our tertiary centre in England achieved a genetic diagnosis in over 40% of cases of selected fetal structural anomaly, following multidisciplinary case reviews and identification of cases with likely monogenic aetiology for further testing [16].

Genetic counselling in prenatal diagnosis is a detailed and complex area and parents require lengthy pre- and post- test counselling. In the UK, counselling surrounding prenatal tests is often undertaken by screening midwives and specialist genetics counsellors, and invasive prenatal testing then discussed by the Fetal Medicine team. Combining the two areas of expertise into a joint clinic can offer parents the best experience for these discussions. Parents often need repetition of statements and additional time for questions to be answered compared with conventional antenatal counselling [17,18,19]. They are also required to have a high level of understanding of the genetic tests and the possible results, far beyond the discussions and decisions to be made with uncomplicated pregnancies, to gain valid consent.

Conventional genetic testing, for a fetus identified with a congenital abnormality on ultrasound scanning, usually requires invasive sampling to obtain an amniocentesis or chorionic villus sample. Genetic testing may include a rapid test for common aneuploidies by quantitative fluorescence-polymerase chain reaction (QF-PCR), and then chromosomal microarray (CMA) or single nucleotide polymorphism (SNP) array to detect unbalanced chromosome rearrangements including copy number variants (CNVs). We know that pathogenic unbalanced chromosomal rearrangements are detected in an additional 5% of fetuses with ultrasound differences with traditional karyotyping, of those around 30% will have autosomal trisomy [20], with array analysis increasing the diagnostic yield by an additional 4.1% [21]. In many centres, karyotype has now been superseded by upfront array analysis, with an overall detection rate of chromosome imbalance/CNVs in 7–10% of pregnancies [22].

## 4. Cell Free Fetal DNA (cffDNA) Based Testing

Non-invasive prenatal testing (NIPT) and non-invasive prenatal diagnosis (NIPD) using cell free fetal DNA from maternal blood significantly reduces the need for invasive diagnostic testing. NIPT is widely used in the UK as a contingent screening test for aneuploidy, where standard first trimester screening results in high chance of a common autosomal trisomy. NIPT cannot be used for diagnostic testing in this context due to the impact of placental mosaicism, and therefore for “high chance” results there is a requirement for an invasive test for confirmation. Non-invasive prenatal diagnosis (NIPD) has been introduced for specific tests, for example to ascertain fetal rhesus status, allowing a streamlined approach to administering anti-D prophylaxis to rhesus negative women with a rhesus positive fetus at risk of hemolytic disease of the newborn. It is also used for fetal sex determination where the pregnancy is at risk of a sex-linked or sex-limited disorder, or where there are ambiguous genitalia on scan. Additionally, since 2012 the NHS has supported the routine use of NIPD for the definitive diagnosis of achondroplasia and thanatophoric dysplasia [23], when suspected on ultrasound, when these arise de novo in the pregnancy. Technically it is relatively straightforward to detect a pathogenic SNV that is not present in the mother. Therefore, it is possible to offer NIPD for de novo or paternally inherited variants, or for exclusion of the paternal variant in an autosomal recessive disorder where both parents carry different pathogenic variants. Since the advent of next generation sequencing and digital PCR, the potential of cell-free fetal DNA based testing for monogenic disorders has increased dramatically. There are a number of reports in the literature reporting routine use for pregnancies where there is a relevant family history of a specific disorder [24,25,26,27]. There is ongoing translational research into circulating fetal cells and circulating fetal trophoblast cells (CFTCs) for NIPD of monogenic disease [28,29]. CFTCs could potentially be used for diagnosis of triplet repeat expansions or point mutations, variants which cffDNA cannot pick up. If validated testing procedures can be developed to give rapid and complete genomic information during pregnancy, the risks of invasive testing could be avoided.

Invasive testing currently remains the gold standard for diagnostic genetic testing for monogenic disorders where testing is carried out due to an abnormal ultrasound scan.

## 5. Next Generation Sequencing: Techniques Used for Prenatal Diagnosis

### 5.1. Exome Sequencing

The most commonly used method of next generation sequencing in the prenatal setting is exome sequencing (ES). This evaluates the protein-coding exons, which although comprise only around 1–2% of the genome, are responsible for more than 85% of all pathological variants which cause disease [30]. ES is often the preferred method of NGS used in clinical practice [2,3] as adopted by the NHS in the R21 and R14 pathways. Although a compromise in some respects as it does not evaluate the whole genome and therefore limits the potential for making a genetic diagnosis, it can detect single nucleotide variants (SNVs), small insertions and deletions and some copy number variants [20] at a relatively lower cost whilst still achieving significant diagnostic yield with a rapid turnaround time, and reducing the discovery of incidental findings. The studies included in this review evaluate whole exome sequencing, or exome sequencing (ES) rather than clinical or targeted exome sequencing. However, there may be variable approaches to analysis, with some analysing the whole exome, and others only analysing a limited panel of genes.

### 5.2. Whole Genome Sequencing

Whole genome sequencing (WGS) is more comprehensive than WES as it involves sequencing the entire genome (all exons and intronic regions of DNA). As it has the potential to identify non-coding pathogenic variants [31], its likely diagnostic yield is greater than ES. The bigger challenge lies in interpreting the clinical significance of any non-coding variants. As the cost and turnaround time of WGS decreases, with technological and bioinformatic pathway advances, and our understanding of the non-coding region of the genome improves, it has the potential to supersede ES for enhanced diagnostic genetic testing in cases of fetal malformations. Although pES is the predominant approach for prenatal genetic testing, small series of prenatal whole genome sequencing are now emerging and there are ongoing trials [32]. The use of genome-wide sequencing was recently covered by an updated position statement by the International Society for Prenatal Diagnosis (ISPD) [33].

## 6. Variant Classification and Challenges Faced

As there are increasing novel sequence variants being detected by laboratories, in 2015 the American College of Medical Genetics and Genomics (ACMG) released guidance on the classification of variants [34]. The understanding that the clinical significance of a variant lies on a spectrum from almost certainly pathogenic to benign is reflected by the update. Variants are classified as either pathogenic, likely pathogenic, uncertain significance, likely benign or benign according to a series of criteria with levels of evidence to support this. Their consensus is that a variant is likely pathogenic/pathogenic or likely benign/benign when there is more than 90% certainty that the variant is causative or not causative of the disease. Further development of the guidelines is constantly undertaken through the US ClinGen Sequence variant interpretation (SVI) working group [35]. The Association for Clinical Genomic Science (ACGS) in the UK recommended the adoption of the guidelines and published their own additional recommendations to facilitate this [36].

As evidence for the pathogenicity and clinical significance evolves, some variants originally classified as uncertain significance may need to be re-classified. In the context of prenatal ES, this is particularly evident as often the fetal phenotype evolves as the pregnancy progresses or where the postnatal phenotype then supports the diagnosis. This is demonstrated by our group’s prospective cohort study: ‘Evolving fetal phenotypes and the clinical impact of progressive prenatal exome sequencing pathways’ [4], which evaluated the R21 pathway. In fetuses with a causative pathogenic variant, 73.3% had additional anomalies diagnosed as the pregnancy progressed. Additionally, three cases were reported where information based on the postnatal phenotype reclassified variants of uncertain significance (VUS) to likely pathogenic. Another recent study ‘Lessons learned from prenatal exome sequencing’ by Chandler et al. found that 20.9% (24 out of 113) of cases were identified as challenging to interpret and report [6]. Three variants in this study were reclassified from VUS to likely pathogenic due to further information from a diagnostic laboratory, an evolving fetal phenotype and new data published in the literature. Postnatal examination of another case revealed additional features, which were consistent with the pathogenic variant identified confirming the diagnosis. Other challenges encountered by Chandler et al. were autosomal dominant variants identified in parental DNA that would otherwise have been missed with inheritance filtering as the parents were seemingly unaffected [6]. Inheritance filtering can be used to expedite reporting by reducing the number of variants sequenced, however, can lead to missed diagnoses in conditions with variable phenotype or decreased penetrance. It is important to maintain close communication with the referring clinician and clinical scientist as fetal phenotypes evolve and new evidence is published in the literature, which may affect variant classification.

## 7. Diagnostic Yield of Next Generation Sequencing in Single System and Multisystem Fetal Disease

### 7.1. Cardiac

An important increase in diagnostic yield from NGS compared to conventional testing has been demonstrated in fetuses with congenital malformations, particularly with multisystem differences where the likelihood of monogenic aetiology is increased [2,3]. To determine its clinical utility in single system disease, our group performed a prospective cohort study [37], using an extended cohort from the PAGE study [2] and systematic review, which included the complete data set from Petrovski et al. [3], evaluating the diagnostic yield of exome sequencing for congenital heart disease (CHD). The additional yield for ES in all CHD was 12.7% (25 out of 197), in isolated CHD this was 11.5% (14 out of 122) and where CHD was associated with extra-cardiac anomalies (ECA) the additional yield was 14.7% (11 out of 75). The pooled incremental yields from all 18 of the included studies were: 21% (95% CI 15–27%, *p* < 0.00001) for all CHD, 11% (95% CI 7–15%, *p* < 0.00001) for isolated CHD and 37% (95% CI 18–56%, *p* = 0.0001) for CHD with ECA. The most common genetic syndromes identified were Kabuki syndrome (19.8%), CHARGE (Coloboma-Heart defects-Atresia choanae-Retardation of growth-Genital abnormalities-Ear abnormalities) syndrome (8.3%), Noonan syndrome (6.3%) and primary ciliary dyskinesia (6.3%).

Two cohort studies have been published since the CODE study. Qiao et al. [38] report data from 360 unselected singleton pregnancies with CHD diagnosed on fetal echocardiography, who were referred for CMA and then ES if CMA was non-diagnostic. Exome sequencing was informative in 6.7% of cases. In a cohort of 260 fetuses with CHD, with a negative karyotype and CMA analysis, Li et al. [39] reported an additional diagnostic yield of 10% (26 out of 260) from WES.

### 7.2. Neurological

Both the PAGE and Petrovski studies reported that ES supported a diagnosis in 8–10% of cases of fetuses with central nervous system (CNS) malformations [2,3], however the full details of these conditions were not published. In response to this, Baptiste et al. [40] performed an expanded review of the respective cohorts to determine the incremental yield of NGS for isolated CNS anomalies, complex CNS anomalies and CNS anomalies associated with malformations in other system(s). 268 fetuses were sequenced, and of those 13.8% (37 out of 268) were found to have a pathogenic or likely pathogenic variant. A causative pathogenic or likely pathogenic variant was found in 7.2% (7 out of 97) of those with an isolated, single CNS anomaly, of which the most common was isolated mild ventriculomegaly. The pathogenic variants for isolated mild ventriculomegaly were found in the CHD7, B3GLCT and ARID1A genes. The CHD7 gene variant was de novo variant and is associated with CHARGE syndrome. B3GLCT gene variants are recessive and cause Peters syndrome, and the ARID1A variant also occurred de novo. Of fetuses with isolated agenesis of the corpus callosum, 30% had a pathogenic variant, in the L1CAM, SHH and PTCH1 genes. 19% (12 out of 63) of fetuses with multiple CNS anomalies were found to have pathogenic or likely pathogenic genetic variants and in the 108 cases with anomalies in multiple organ systems, 16.7% (18 fetuses) had causative pathogenic or likely pathogenic variants. The most likely fetuses to have a positive diagnosis were those with CNS and renal or genitourinary anomalies. In terms of inheritance, 54% (20 out of 37) were inherited from one or both parents: 17 autosomal recessive, one autosomal dominant and one was X-linked recessive. De novo mutations were found in 46% (17 out of 37): 15 of these were autosomal dominant, 2 were X-linked dominant.

This data supports the use of ES as a secondary diagnostic genetic test for CNS anomalies, particularly with the finding of 13% of fetuses with mild isolated ventriculomegaly having a causative pathogenic or likely pathogenic variant found on ES. This highlights the importance of offering NGS as a secondary test in isolated ventriculomegaly to provide parents with reassurance, rather than relying on standard genetic testing alone. It is important to reach a consensus as to what constitutes a ‘major CNS malformation’, particularly as evidence emerges for incremental diagnostic yield of exome sequencing with less severe neurological abnormalities.

### 7.3. Increased Nuchal Translucency

Another review by Mellis et al. [41] used the final extended data sets from the PAGE [2] and Petrovski [3] studies to examine fetuses with an increased nuchal translucency (NT) of at least 3.5 mm, to identify whether these cases would also benefit from prenatal ES. In total, 213 fetuses were identified with increased NT, 159 were initially classified as isolated and 54 were associated with other structural anomalies. Altogether, 13.1% (28 out of 213) of cases had a causative variant following ES. Of these, 22.2% (12 out of 54) of fetuses with increased NT and other anomalies had a causative variant, whereas only 1.8% of those with an isolated increased NT (which remained isolated throughout) had a diagnostic variant detected. They found a significant increase in the diagnostic rate with fetuses initially presenting with increased NT with additional anomalies (1.8% versus 22.2%. *p* < 0.001) and those where additional abnormalities were discovered later (1.8% versus 32.4%, *p* < 0.001), compared to those with isolated increased NT. Additionally, the diagnostic rate increased from 1.6% with NT 3.5–4.4 mm to 28.6% with NT > 7.5 mm (*p* < 0.05). The results are in line with existing evidence in isolated increased NT that if array analysis is normal, the increased NT resolves and no other abnormalities are found later in pregnancy, then parents can be reassured that it is very likely they will have a healthy infant without any major abnormalities [42,43]. However, it raises the question as to whether isolated increased NT > 7.5 mm should be offered ES testing if karyotype and CMA are normal, as in this small series 28.6% (4 out of 14) of fetuses had a pathogenic variant. Currently, we are constrained by capacity and the worry of identifying variants of uncertain significance to offer ES for all fetuses with isolated NT within the NHS system, however this may change in the future.

In pregnancies with increased NT with other anomalies, Noonan syndrome was diagnosed in 33.3% (4 out of 12) of fetuses, and a further 6 fetuses had variants in Noonan syndrome genes that were deemed ‘potentially clinically relevant’. For the two cases of isolated increased NT, one was diagnosed with maternal chromosome 15 uniparental disomy, not found on CMA and the other was found to have a de novo frameshift variant in the *RERE* gene. At birth this infant did not have any features consistent with *RERE* disease but developed clinical signs at around 8 months of age. It is notable that their cohort included three diagnoses of Noonan syndrome, with the causative variant being inherited from undiagnosed affected parents. Two families had a history of previous pregnancy loss with a clinical phenotype consistent with Noonan syndrome and in two cases the affected parent had unrecognized, but clinical features consistent with this diagnosis. This highlights the importance of taking thorough obstetric and family histories, as well as careful parental examination to guide a molecular diagnosis, particularly if the suspected genes display variable expression or penetrance.

Overall, this paper suggests that offering NGS as a secondary genetic test is clinically useful in fetuses with increased NT with other anomalies and could be useful in those isolated cases if the NT is greater than 7.5 mm.

### 7.4. Hydrops

Our group performed a prospective cohort study and meta-analysis to determine the additional incremental yield of NGS in fetuses diagnosed prenatally with non-immune hydrops fetalis (NIHF), with normal karyotype or CMA analysis [44]. This further contributes to the evidence base for NGS in single system congenital malformations. The cohort was an extended data set from the PAGE study [2], and a meta-analysis of 21 studies was also performed, including the study by Petrovksi et al. [3], to include a total of 306 cases. The pooled incremental yield of ES in all cases of NIHF was 29% (95% CI 24–34%, *p* < 0.00001), in those with isolated NIHF it was 21% (95% CI 13–30%, *p* < 0.00001) and in those with NIHF with additional structural anomalies it was 39% (95% CI 30–49%, *p* < 0.00001). The genetic disorders most commonly found were RASopathies in 30.3% (27 out of 89) of cases, with a variant in the PTPN11 gene most common, musculoskeletal disorders in 14.6% (13 out of 89) of cases, due to RYR1 variants in 38.5% (5 out of 13), and inborn errors of metabolism in 12.4% (11 out of 89), of which GUSB variants accounted for 45.5% (5 out of 11). When a genetic cause for hydrops was established, a dominant condition was most common, but the majority of variants occurred de novo.

This systematic review demonstrates that when pES is used as a secondary genetic test for fetuses with prenatal NIHF, the additional diagnostic yield is 29%, when karyotyping and/or CMA is non-diagnostic. Even in those fetuses with isolated NIHF there was significant diagnostic yield, with a RASopathy the most common diagnosis. This data supports its clinical utility in all cases of NIHF diagnosed prenatally. As a result of this study, and the corroboration of similar findings in other smaller studies, NIHF was included in the NHS England R21 pathway from March 2021 [4,5].

### 7.5. Renal and Urinary Tract

Whilst there have been no large cohort studies or meta-analyses of the incremental yield of NGS with disorders of the renal and urinary tract, there are two small cohort studies from China which have assessed the additional diagnostic yield of whole exome sequencing in fetuses with congenital anomalies of the kidney and urinary tract (CAKUT), where the karyotype and CMA are normal. Zhou et al. [45] examined the WES of 41 fetuses with isolated unexplained CAKUT. The detection rate for pathogenic variants was 7.32% (3 out of 41), with incidental variants detected in 2.4% (1 out of 41). All the fetuses with a pathogenic variant had bilateral CAKUT. Lei et al. [46] reviewed 30 cases of CAKUT with a normal karyotype and normal CMA and found 13% (4 out of 30) had a pathogenic variant and 7% (2 out 30) had an incidental variant following WES. Follow-up studies are needed to clearly demonstrate clinical utility for NGS in CAKUT. However, these results suggest it is worthwhile, particularly in bilateral abnormalities.

### 7.6. Skeletal Dysplasias

The incremental yield of next generation sequencing in skeletal dysplasia has not yet been examined from a large data set. However, a recent small cohort study by Zhang et al. [47] reviewed 55 fetuses with skeletal dysplasia suspected on ultrasound scan, with a normal karyotype and CMA, who had WES performed as a secondary genetic test. Their results showed a diagnostic yield of 64% (35 out of 55), with 37 different pathogenic or likely pathogenic variants; in 14 cases these variants were de novo. Kucinska-Chahwan et al. [48] observed a total cohort of 55 cases of skeletal dysplasia. ES was performed on 26 out of 55 fetuses, where karyotyping and CMA was not diagnostic, and found an additional diagnostic yield of 32.7%. 69.2% (18 out of 26) received a diagnosis after ES. Pathogenic or likely pathogenic variants were found in 14 different genes and the mode of inheritance was autosomal dominant in 12 cases and autosomal recessive in 6 cases. Another paper of 38 cases demonstrated pathogenic or likely pathogenic variants in 65.79% (25 out of 38) cases of fetal skeletal dysplasia detected on ultrasound, using WES [49]. This study found 28 variants affecting 10 genes, of which 35.71% (10 out of 28) were novel. Several small cohort studies demonstrate the need for further research. However, these studies show a high additional diagnostic yield of NGS as a secondary genetic test. This suggests its clinical utility for fetuses with skeletal dysplasia on ultrasound when karyotype and CMA are normal.

### 7.7. Overall Diagnostic Yield of NGS in Single System Congenital Abnormalities

A recent systematic review and meta-analysis published by Mellis et al., collates all the data published to date regarding the diagnostic yield of exome sequencing for prenatal diagnosis of fetal structural anomalies [5]. They reviewed 72 reports from 66 studies, which includes data from 4350 fetuses in total. Their pooled incremental yield of exome sequencing for all anomalies (both multi- and single system) was 31% (95% CI 26–36%, *p* < 0.0001), which was their primary outcome. They also performed sub-group analyses on specific fetal phenotypes to determine the incremental yield of ES for single system congenital abnormalities. Their results showed that diagnostic yield varied significantly according to fetal phenotype. Their pooled estimated diagnostic yield for each isolated system was as follows: skeletal 53% (95% CI 42–63%, *p* < 0.0001), neuromuscular/fetal akinesia deformation sequence 37% (95% CI 20–54%, *p* < 0.0001), multisystem 29% (95% CI 22–35%, *p* < 0.0001), hydrops/oedema 22% (95% CI 14–31%, *p* < 0.0001), central nervous system (CNS) 17% (95% CI 12–22%, *p* < 0.0001), cardiac 11% (95% CI 7–16%, *p* < 0.0001), craniofacial 9% (95% CI 1–17%, *p* = 0.02), congenital anomalies of the kidneys and urinary tract (CAKUT) 9% (95% CI 5–12%, *p* < 0.0001), fetal growth restriction (FGR) 4% (95% CI −9–17%, *p* = 0.59), isolated increased nuchal translucency (NT) 2% (95% CI 0–5%, *p* = 0.04), gastrointestinal 2% (95% CI −4–8%, *p* = 0.5), respiratory/chest 0 (95% CI −7–7%, *p* = 1), abdominal wall 0 (95% CI −31–31%, *p* = 1). We have created a forest plot using this data to summarise the diagnostic yield for each system (Figure 1). There was great variation in sample size for the sub-group analyses of fetal phenotype, and the samples for fetuses with gastrointestinal, respiratory and chest abnormalities, abdominal wall defects and fetal growth restriction were too small to demonstrate statistical significance.

To compare these results with the diagnostic yield demonstrated in the post-PAGE and Columbia cohorts discussed in detail above, for cardiac disease our group observed an incremental yield of 12.7%, 13.8% for neurological abnormalities, 13.1% in fetuses with increased nuchal translucency and 29% for non-immune fetal hydrops. This is relatively consistent with yield observed in this systematic review, except from their yield for increased nuchal translucency of only 2%. This could be explained by substantial heterogeneity between the included studies in this subgroup, which varied in sample size, their approach to interpretation of the fetal phenotype (prenatal alone or with postnatal/postmortem results), variant interpretation and their sequencing and analysis approach.

## 8. Genetic and Genomic Investigation in the Neonatal Period

Next generation sequencing has been shown to be clinically useful in neonatal and paediatric intensive care units for the diagnosis of suspected monogenic disorders in unwell infants, where prenatal testing has not occurred. It is particularly pertinent in the critically unwell infant and guides redirection of care, allows the opportunity for further specialist input, facilitates changes in medication or therapy and can provide information for parents regarding prognosis and recurrence risk [9]. As a result of its clear utility in this clinical setting the NHS in England introduced the R14 pathway, using the National Genomic Test Directory [7], to employ rapid trio exome sequencing with a swift turnaround time to allow for timely diagnosis and appropriate changes in clinical management.

In addition to the diagnosis of monogenic disorders in the unwell neonate, there is also argument for using NGS as part of routine newborn screening. Although there is not a consensus agreement as to its utility in this context, the aim would be to expand and enhance the diagnosis of rare monogenic and metabolic disorders that can affect infants and children, even if not clinically apparent in the neonatal period. The BabySeq Project [50] in the United States of America is an RCT designed to evaluate the medical, behavioural and economic impact of whole genome sequencing in two cohorts of healthy and sick neonates, compared with control cohorts of neonate who receive standard screening and care. However, it is of concern that sequencing and storing an individual’s whole genome may unearth findings that may not be relevant initially but would impact in later life, such as the presence of pathogenic variants in cancer causing genes, or inherited diseases that do not present until later life. It also poses the question of data storage and security and how this information could be used inappropriately in the future. Before considering the potential use of WGS as part of newborn screening within the NHS, the UK National Screening Committee and Genomics England commissioned a public dialogue in September 2020 to ascertain the UK public’s views on this [51]. The final report reveals that participants were overall in support of WGS as part of the newborn screening program, particularly if it was used to identify a wider set of conditions that could impact the infant in early childhood, which have interventions or treatments to prevent, cure or slow progression. They acknowledge that the consent process for WGS should take into account complexities such as: its implications on the wider family, that the family structure can come in many forms, that while the parents may give consent the child may have differing views as they mature and that the screening test could potentially identify many more conditions than current newborn screening tests.

## 9. Postmortem Clinico-Pathological Correlation

In pregnancies with fetuses who have a structural abnormality, that result in termination of pregnancy, intrauterine fetal death or neonatal death next generation sequencing can be used as a secondary genetic test at postmortem examination, when standard karyotyping and CMA is non-diagnostic. We examined a cohort of 27 structurally abnormal fetuses undergoing postmortem examination, whereby QF-PCR and CMA were normal and found a diagnostic yield from trio exome sequencing of 37% (10 out of 27 with pathogenic or likely pathogenic variant) [52]. Four of these variants had arisen de novo, with the 6 remaining variants being inherited (autosomal recessive or X-linked)). A previous study by Yates et al. [53] reviewed a cohort of 84 fetuses with structural abnormalities having postmortem investigation with whole exome sequencing and revealed a diagnostic yield of 20% (17 out of 84). The comparatively increased yield found in a study from our group is likely explained by the use of detailed postmortem findings in conjunction with imaging features to determine the fetal phenotype in a multidisciplinary group. Yates et al. describe the limitation of using ultrasound findings alone, with variation in the amount of detail provided in scan reports. Postmortem examination can detect subtle dysmorphic features, not visualised on ultrasound, which allows the fetal phenotype to be determined more accurately.

Both studies show the clinical utility of using NGS as an additional diagnostic test at postmortem examination in the absence of an antenatal diagnosis, as establishing a diagnosis aids accurate genetic counselling for parents, which will include accurate recurrence risks and planning for future pregnancies.

## 10. Planning a Future Pregnancy

Once the diagnosis of a monogenic disorder has been made in a fetus with congenital abnormalities, the knowledge of this can guide planning of future pregnancies for the parents. Comprehensive genetic counselling is vital for understanding inheritance patterns, and risk of recurrence in subsequent pregnancies, which can be as high as a 1:2 or 1:4 risk. This counselling may be best delivered through a multi-disciplinary clinic including screening midwives, specialist genetics counsellors and the Fetal Medicine team. Parents can be offered the option of preimplantation genetic testing (PGT) via in vitro fertilization (IVF) techniques prior to pregnancy. Alternatively, prenatal testing can be offered during the pregnancy, either through invasive testing or non-invasive testing (NIPD) dependent on the genetic abnormality. All of these approaches are increasingly using next generation sequencing as a testing methodology [54]. Early detailed ultrasound scanning by a Fetal Medicine Specialist can also be carried out alongside the genetic testing, however fetal structural abnormalities associated with the disorder may not be evident until later in the pregnancy. Testing of a CVS sample taken at 11–14 weeks of pregnancy, or a maternal blood sample taken for cffDNA testing from 8 weeks of pregnancy, can usually give a definitive result much earlier in the pregnancy as to whether or not the fetus is affected. This allows for more time for counselling and management decisions.

## 11. Conclusions

Next generation sequencing is increasingly used for the diagnosis of monogenic disorders in fetuses with structural congenital abnormalities, as demonstrated by the R21 pathway in the NHS in England, as part of postnatal investigation of the sick neonate via the R14 pathway, and in the R27 and R412 pathways in pregnancy loss. There is clear evidence to support the use of exome sequencing in multi-system congenital abnormalities, however does it offer additional yield in single system disease? The evidence from the cohorts that have followed the PAGE Study and Petrovski et al. demonstrate significant incremental diagnostic yield of exome sequencing as a secondary diagnostic genetic test for single system congenital abnormalities, with particular focus on cardiac and neurological anomalies, increased nuchal translucency and non-immune fetal hydrops [29,32,33,36]. This supports its translation into routine clinical practice in Fetal Medicine departments for the diagnosis of monogenic conditions in fetuses with both multi- and single system abnormalities.

There is also great potential for next generation sequencing to be utilised in non-invasive prenatal diagnosis as technology advances, which avoids the risks of invasive testing, and also as a screening tool for the healthy newborn. It can be used as second line genetic testing in post mortem examination if prenatal testing is not performed and can aid future pregnancy planning using pre-implantation genetic diagnosis via IVF treatment.

Next generation sequencing offers a wide spectrum of benefits to families in the diagnosis and management of monogenic disease and will significantly advance practice in fetal medicine, reproductive medicine, neonatology and clinical genetics worldwide.

## Figures and Tables

**Figure 1 genes-13-01517-f001:**
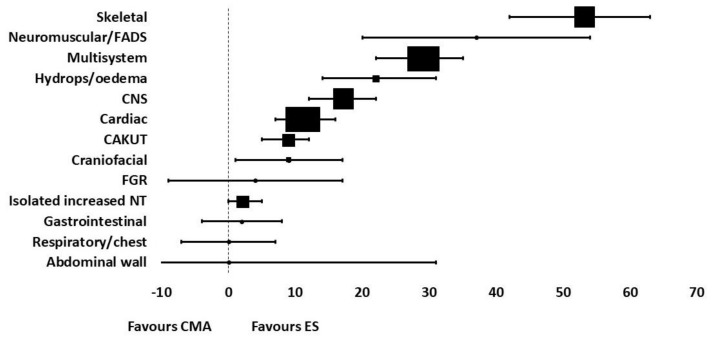
Forest plot summarizing the diagnostic yield of exome sequencing for abnormalities in each system, using data from the systematic review by Mellis et al. [5].

## Data Availability

No new data were created or analyzed in this study. Data sharing is not applicable to this article.

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
