# Peer review of "Next Generation Sequencing after Invasive Prenatal Testing in Fetuses with Congenital Malformations: Prenatal or Neonatal Investigation"

_genes, 2022, doi:10.3390/genes13091517_

Round 1
Reviewer 1 Report
Summary
The authors review several studies that have applied NGS in cases of congenital malformation detected by ultrasound screening. The review focuses specifically on single-system anomalies vs the initial unselected prospective cohort studies. The authors highlight the increased yield when focus is narrowed to specific single-system anomalies and they provide sound evidence for the use of NGS in routine clinical care in cases of ultrasound detection of a single-system congenital malformation.
General concept comments
The review is very thorough and provides sufficient background information on the studies included. As NGS is increasingly being used clinically, knowledge of the utility for specific single-system anomalies is particularly valuable.
Specific comments
Lines 121 – 131 – These are confusing, consider rewording. “detection rate of autosomal trisomy is around 30%” I see by looking at the reference that 5% of pregnancies have structural abnormality and of those 30% will have trisomy. However, the way it reads in this review 30% could be confused for the sensitivity for trisomy detection. Also, “karyotype analysis is now rarely used… additional 5% of fetuses with ultrasound differences with traditional karyotyping.” I see what you are getting at, but the way it reads is confusing.
Line 210 – 24 out of 113 is not 11.3%. 11.3% were just analysis/reporting troubles.
Line 260 - B3GLT should be B3GLCT.
Line 286 – Since it was isolated at the time of ES, 1.8% doesn’t seem like a meaningful number here. 16 of the 159 (10%) that were isolated at the time of ES had diagnostic variants identified. This still shows it is lower than when multiple anomalies are identified, but doesn’t underestimate the potential clinical benefit of ES in the same way 1.8% does; the 12 with diagnostic variants identified that developed additional anomalies still benefited from ES at the time increased NT triggered ES.
Author Response
We appreciate the valuable comments of the reviewers and are grateful to the editors for the opportunity to have our revised work considered for publication. A number of changes have helped us to improve the paper for your readership and we hope that you find it worthy of inclusion in Genes.
Summary
The authors review several studies that have applied NGS in cases of congenital malformation detected by ultrasound screening. The review focuses specifically on single-system anomalies vs the initial unselected prospective cohort studies. The authors highlight the increased yield when focus is narrowed to specific single-system anomalies and they provide sound evidence for the use of NGS in routine clinical care in cases of ultrasound detection of a single-system congenital malformation.
General concept comments
The review is very thorough and provides sufficient background information on the studies included. As NGS is increasingly being used clinically, knowledge of the utility for specific single-system anomalies is particularly valuable.
Thank you very much for this very positive feedback.
Specific comments
Lines 121 – 131 – These are confusing, consider rewording. “detection rate of autosomal trisomy is around 30%” I see by looking at the reference that 5% of pregnancies have structural abnormality and of those 30% will have trisomy. However, the way it reads in this review 30% could be confused for the sensitivity for trisomy detection. Also, “karyotype analysis is now rarely used… additional 5% of fetuses with ultrasound differences with traditional karyotyping.” I see what you are getting at, but the way it reads is confusing.
Thank you for highlighting this paragraph. We have now reworded it on lines 127-132 to read: “We know that pathogenic unbalanced chromosomal rearrangements are detected in an additional 5% of fetuses with ultrasound differences with traditional karyotyping, of those around 30% will have autosomal trisomy [20], with array analysis increasing the diagnostic yield by an additional 4.1% [21]. In many centres, karyotype has now been superseded by upfront array analysis, with an overall detection rate of chromosome imbalance/CNVs in 7–10% of pregnancies [22].” We hope that this now reads better and is less confusing.
Line 210 – 24 out of 113 is not 11.3%. 11.3% were just analysis/reporting troubles.
Thank you, we have amended this to 20.9% as per the original paper.
Line 260 - B3GLT should be B3GLCT.
Thank you, this typographical error has now been corrected.
Line 286 – Since it was isolated at the time of ES, 1.8% doesn’t seem like a meaningful number here. 16 of the 159 (10%) that were isolated at the time of ES had diagnostic variants identified. This still shows it is lower than when multiple anomalies are identified, but doesn’t underestimate the potential clinical benefit of ES in the same way 1.8% does; the 12 with diagnostic variants identified that developed additional anomalies still benefited from ES at the time increased NT triggered ES.
Thank you for this comment. We agree that this is a valid point and it is all down to when ES is offered and there are two arguments, either you offer to all isolated increased NT at the start, or you wait until other features develop, but then this is much later. This depends on your healthcare system, money available, impact of testing, amongst other factors. We feel that the Mellis paper is probably written from the point of view of the NHS. We agree that for counselling patients at the start of their pregnancy then quoting 1.8% is not relevant, so we have qualified this by adding this statement in brackets: “Altogether, 13.1% (28 out of 213) of cases had a causative variant following ES. Of these, 22.2% (12 out of 54) of fetuses with increased NT and other anomalies had a causative variant, whereas only 1.8% of those with an isolated increased NT (which remained isolated throughout) had a diagnostic variant detected.”. We have also added this sentence at the end of the paragraph: “Currently, we are constrained by capacity and the worry of identifying variants of un-certain significance to offer ES for all fetuses with isolated NT within the NHS system, however this may change in the future.” This hopefully acknowledges that although 1.8% is low, this is still 1 in 55 which is a significant number of genetic diagnoses in this group of patients.
Reviewer 2 Report
This manuscript presents an updated review of the principal publications on NGS application in invasive prenatal diagnosis of cohorts with congenital malformations. Due to the paucity of studies in which WGS is applied on cohorts and prenatally, the review is mainly focused on whole exome sequencing application. Starting principally from the results of 2 large-cohort studies, the paper analyses several classes of single-system malformations (cardiac, neurological, skeletal, renal) and single anomalies (increased nuchal translucency, hydrops), reporting the incremental diagnostic yield obtained when whole exome sequencing is applied in cases with normal karyotype and microarray. The authors concluded supporting the use of whole exome sequencing in routine prenatal care in fetuses with single-system congenital anomalies.
Limitations of the studies discussed and critical points of the diagnostic pathway application and results interpretation are correctly reported. The state-of-art is presented quite carefully. The number and type of cited references are appropriate and the paper is quite well structured.
Minor comments are as follows:
1) Please specify in the text if “ES” is always WES, or in some studies clinical exome sequencing has been applied
2) I suggest inserting a table to summarize the WES diagnostic yield obtained for the distinct classes of malformations in the different studies
3) I suggest removing the not needed self-citations from the references
4) Line 176, please correct “exomes” to “exons”
5) Line 278, I suggest substituting “brain differences” with a more appropriate term
Author Response
We appreciate the valuable comments of the reviewers and are grateful to the editors for the opportunity to have our revised work considered for publication. A number of changes have helped us to improve the paper for your readership and we hope that you find it worthy of inclusion in Genes.
Reviewer comments
This manuscript presents an updated review of the principal publications on NGS application in invasive prenatal diagnosis of cohorts with congenital malformations. Due to the paucity of studies in which WGS is applied on cohorts and prenatally, the review is mainly focused on whole exome sequencing application. Starting principally from the results of 2 large-cohort studies, the paper analyses several classes of single-system malformations (cardiac, neurological, skeletal, renal) and single anomalies (increased nuchal translucency, hydrops), reporting the incremental diagnostic yield obtained when whole exome sequencing is applied in cases with normal karyotype and microarray. The authors concluded supporting the use of whole exome sequencing in routine prenatal care in fetuses with single-system congenital anomalies.
Limitations of the studies discussed and critical points of the diagnostic pathway application and results interpretation are correctly reported. The state-of-art is presented quite carefully. The number and type of cited references are appropriate and the paper is quite well structured.
Thank you very much for this positive feedback.
Minor comments are as follows:
- Please specify in the text if “ES” is always WES, or in some studies clinical exome sequencing has been applied
The studies included in this review evaluate whole exome sequencing, or exome sequencing (ES) rather than clinical or targeted exome sequencing. However, there may be variable approaches to analysis, with some analysing the whole exome, and others only analysing a limited panel of genes. We have included this statement in the text, starting on line 174
- I suggest inserting a table to summarize the WES diagnostic yield obtained for the distinct classes of malformations in the different studies
We have now included a forest plot created from the data published in Mellis et al’s systematic review ‘Diagnostic yield of exome sequencing for prenatal diagnosis of fetal structural anomalies: A systematic review and meta-analysis’ which summarizes the diagnostic yield of WES for each system, I hope that this will suffice? If it is not sufficient we would be grateful for more details of what is suggested, thank you.
- I suggest removing the not needed self-citations from the references
We feel that it is important for the readers to know our references to enable them to look them up. It would be helpful to know which references are thought to be superfluous. Without this information, it's difficult to know which ones to excluded, thank you.
- Line 176, please correct “exomes” to “exons”
This has now been corrected, thank you.
- Line 278, I suggest substituting “brain differences” with a more appropriate term
We have changed this now to “neurological abnormalities”, we hope that you find this a more appropriate term.